# Using the Maize Nested Association Mapping (NAM) Population to Partition Arbuscular Mycorrhizal Effects on Drought Stress Tolerance into Hormonal and Hydraulic Components

**DOI:** 10.3390/ijms23179822

**Published:** 2022-08-29

**Authors:** Juan Manuel Ruiz-Lozano, Gabriela Quiroga, Gorka Erice, Jacob Pérez-Tienda, Ángel María Zamarreño, José María García-Mina, Ricardo Aroca

**Affiliations:** 1Departamento de Microbiología del Suelo y Sistemas Simbióticos, Estación Experimental del Zaidín (CSIC), Profesor Albareda Nº 1, 18008 Granada, Spain; 2Misión Biológica de Galicia (MBG-CSIC), Apartado de correos 28, 36080 Pontevedra, Spain; 3ATENS—Agrotecnologías Naturales S.L., Ctra.T-214, s/n, Km 4, La Riera de Gaia, 43762 Tarragona, Spain; 4Departmento de Biología Ambiental, Facultad de Ciencias, Universidad de Navarra, Irunlarrea No 1, 31008 Pamplona, Spain

**Keywords:** aquaporin, arbuscular mycorrhiza, drought, maize, phytohormone, root hydraulic conductivity

## Abstract

In this study, a first experiment was conducted with the objective of determining how drought stress alters the radial water flow and physiology in the whole maize nested association mapping (NAM) population and to find out which contrasting maize lines should be tested in a second experiment for their responses to drought in combination with an arbuscular mycorrhizal (AM) fungus. Emphasis was placed on determining the role of plant aquaporins and phytohormones in the responses of these contrasting maize lines to cope with drought stress. Results showed that both plant aquaporins and hormones are altered by the AM symbiosis and are highly involved in the physiological responses of maize plants to drought stress. The regulation by the AM symbiosis of aquaporins involved in water transport across cell membranes alters radial water transport in host plants. Hormones such as IAA, SA, ABA and jasmonates must be involved in this process either by regulating the own plant-AM fungus interaction and the activity of aquaporins, or by inducing posttranscriptional changes in these aquaporins, which in turns alter their water transport capacity. An intricate relationship between root hydraulic conductivity, aquaporins and phytohormones has been observed, revealing a complex network controlling water transport in maize roots.

## 1. Introduction

Global warming and reduced rainfall are leading to frequent episodes of drought globally, causing a serious impact on crop growth and productivity [1]. Thus, under the current climate change scenario, increasing crop tolerance to new environmental conditions is of crucial importance in order to secure food production for the ever-increasing human population [2]. In this context, maize (*Zea mays* L.) is one of the most important crops both for animal and human consumption [3]. Maize is cultivated on more than 142 million ha of land worldwide and it is estimated to account for one third of the total global grain production [3]. As a food crop, maize is even more important than other cereals such as rice or wheat [4]. However, maize experiences important decreases in yields in different world regions because it is highly susceptible to drought stress, especially during the reproductive phase [5]. In addition, it has been observed that at the early crop establishment phase, drought stress also influences negatively seedlings adaptation and their potential for grain yield production [4,6]. Hence, there is a need to understand the mechanisms that allow for a higher productivity of crops such as maize under water limiting conditions, which is crucial to guarantee future food production [2,7].

Maize presents an extensive phenotypic and molecular variation [8]. The collection of maize recombinant inbred lines (RILs) in populations is collectively referred to as the “maize Nested Association Mapping (NAM) population” and was developed by the Molecular and Functional Diversity of the Maize Genome project (maizecoop.cropsci.uiuc.edu/nam-rils.php, accessed on 5 July 2022). In this study we have used the inbred parents of the maize NAM RIL population. The RILs are “nested” in the sense that they all share a common parent, B73, but each population has a different alternate parent. The inbred parents of the NAM population cover the full genetic variability in maize [9]. Indeed, more than half are of tropical origin, nine are from temperate areas, two are sweet corn lines and one is a popcorn inbred line. The inbred B73 has been also included as the reference inbred line because of its use for the Maize Sequencing Project (ensembl.gramene.org/Zea_mays/Info/Index, accessed on 5 July 2022). More information can be obtained at: maizecoop.cropsci.uiuc.edu/nam-rils.php (accessed on 5 July 2022) or at panzea.org/#!phenotypes/c1m50 (accessed on 5 July 2022).

There are different strategies to increase crop tolerance to adverse growing conditions, such as the selection of resistant varieties, crossbreeding among varieties, genetic engineering, etc., but modern sustainable agriculture favours use of natural resources such as the soil beneficial microorganisms that can improve plant development under limiting environmental conditions. Among these microorganisms are certain soil fungi that establish a mutualistic symbiosis with the roots of most plants, called mycorrhiza (fungus-root), omnipresent in all biomes and terrestrial ecosystems for more than 400 million years. Among them, the arbuscular mycorrhizal (AM) fungi stand out [10]. The AM symbiosis activates the host’s physiological, molecular and morphological plant responses, increasing the ability of host plants to survive and maintain vigour under drought conditions. Hence, the AM fungi-plant interaction represents a clear example of a sustainable agricultural strategy and represents one of the most sustainable strategies to increase the tolerance of plants to adverse conditions such as water limitation [1,11,12].

The positive effects of the AM symbiosis against drought have been explained not only as a result of a more efficient nutrient uptake and transfer of water through the fungal hyphae to the host plant [13] but also because of changes in soil water retention properties and soil structure, better osmotic adjustment of AM plants, enhancement of plant gas exchange and water use efficiency and protection against the oxidative damage generated by drought reviewed by [1,12,14,15].

Moreover, the AM symbiosis modifies membrane-associated proteins such as aquaporins. These proteins are small channels that allow for the passage of water and small molecules through the membranes of most living organisms. In plants such as maize, they constitute a large family (>30 members) subdivided in four subfamilies: plasma membrane intrinsic proteins (PIPs), tonoplast intrinsic proteins (TIPs), nodulin 26-like intrinsic proteins (NIPs) and small basic intrinsic proteins (SIPs) [16,17]. Aquaporins represent the main pathway for water movement through the cell membranes [17,18], providing the capacity for rapidly modify membrane water permeability, which help the plant with the maintenance of the water balance during stress episodes, and affecting root hydraulic conductivity [19,20]. Furthermore, their relevance for plant physiology is emphasized by the fact that, apart from water, some aquaporin isoforms can facilitate membrane diffusion of other small solutes such as CO_2_, metalloids, urea, ammonia, H_2_O_2_, oxygen or even ions [18,21,22]. Indeed, the importance of aquaporins for both water and solutes exchange during AM symbiosis has been recognized [23] and is supported by more recent results obtained by our research group [6,24,25,26,27]. 

In plants, the development and responses to stress are mainly regulated by a complex hormonal crosstalk including abscisic acid (ABA), jasmonates, salicylic acid (SA) or auxins, [28,29] and this crosstalk has been shown to be altered by the AM symbiosis [30,31]. ABA modulates plant water relations by altering root hydraulic conductance and/or transpiration rate, as well as, by inducing genes encoding for proteins involved in cellular dehydration tolerance [32]. Jasmonic acid (JA) and methyl jasmonate (MeJA) are implicated in plant development and also play a role against abiotic and biotic stresses [33,34]. Salicylic acid (SA) is involved in the regulation of photosynthesis, antioxidant defence system, nitrogen metabolism and plant water relations. Thus, it provides protection against abiotic stresses [35,36]. Auxins regulate root initiation and elongation and have been related to the process of AM fungal colonization. Evidence has shown that auxin signalling is required for normal AM infection [37], for arbuscule development [38] and for plant responses to drought [39,40]. 

Therefore, the general objective of this study was to determine how drought stress alters the capacity for radial water movement and physiology in the whole maize NAM population, in order to find out contrasting maize lines to be tested for their responses to drought in combination with an AM fungus. The role of plant aquaporins and phytohormones in the responses of these contrasting maize lines to cope with drought stress will be unravelled. For that, a group of maize aquaporins previously shown to be regulated by the AM symbiosis under different conditions of drought stress (moderate or severe) and in different varieties of maize (drought-sensitive and drought-tolerant) [6,24] where chosen for analysis of their expression in the selected maize lines.

## 2. Results

### 2.1. Experiment 1

#### 2.1.1. Plant Shoot Dry Weight (SDW)

The highest SDW value was obtained in line Ms71, followed by Mo18W, B73, B97, Tzi8 or CML52 cultivated under well-watered conditions, while the lowest SDW values were obtained in lines Ki3, IL14h, Tx303, NC350, M162W or M37W, regardless of the water regime imposed (Figure 1). In general, the lines with a higher SDW were affected more negatively by drought stress, significantly decreasing this parameter, except Tzi8 and CML52, which maintained a similar SDW under both water regimes. 

#### 2.1.2. Stomatal Conductance (Gs) and Efficiency of Photosystem II

The stomatal conductance decreased significantly in almost all maize lines due to the drought stress imposed, except in CML277, HP301, Ki11, NC358 or CML322 (Figure 2A). In contrast, the efficiency of photosystem II resulted less affected by drought stress and decreased significantly only in lines P39, Mo17, CML152, Ms71 and Tx303 (Figure 2B).

#### 2.1.3. Osmotic Root Hydraulic Conductivity (Lo)

The Lo values showed a very high variability among maize lines (Figure 3). Some of these lines exhibited higher Lo values under drought stress conditions than under well-watered conditions (Tzi8, HP301, Ky21, CML69, Ki11, Mo18W or IL14h). Moreover some of these lines had a high Lo value together with a high shoot dry weight (Mo18W, Ky21 and Tzi8). On the other hand, several maize lines significantly decreased Lo values when subjected to drought stress (B73, NC350, M162W, CML228, CML322, Tx303 or CML333). 

#### 2.1.4. Hydrostatic Root Hydraulic Conductivity (Lh)

In general, the Lh values exhibited less variability and effect of drought stress than Lo values (Figure 4). Thus, few maize lines showed higher Lh values under drought stress than under well-watered conditions (P39, CML52, Ki11, CML228 or Tx303). Even fewer maize lines showed a significant reduction in Lh values due to drought stress (Ki3, B97 or CML103). The rest of maize lines showed no significant variation in Lh values due to the drought stress imposed. 

#### 2.1.5. Principal Component Analysis

We also analysed the sap content in hormones such as ABA, IAA, SA, JA and JA-Ile. The hormones showing a higher response and variability among treatments were ABA and JA-Ile (See Appendix A). We conducted a principal component analysis to disentangle the maize variability in hydraulic, hormonal and growth responses to drought stress (Figure 5) and to select a group of lines with contrasting behaviour for subsequent experiment 2. 

Under well-watered conditions (Figure 5A) most of the variability among maize lines was explained on the basis of physical parameters (PC1 component) such as Lo, Lh, SDW or GS, with a lower influence of the hormonal parameters (PC2 component). Under such conditions, lines Ms71, B97, Tzi8, Ky21 and CML52, exhibited a high dependency of Lh, transpiration (Gs) and Lo values and correlated negatively with all the hormones measured in sap (ABA, IAA, SA, JA and JA-Ile). A group of lines such as Ki11, IL14h or Tx303 were more dependent on SA levels in sap and correlated negatively with jasmonates, Lo, Gs and Lh. Lines Mo18W and B73 depended to a lesser extent on Lo, Lh, Gs or jasmonates and correlated negatively with SA levels in sap. Finally, lines NC350 and NC358 were highly dependent on levels of the hormones (ABA, IAA, JA and JA-Ile) and had low dependency of Lo, Lh, Gs or SA levels. 

Under drought stress conditions (Figure 5B) most of the variability among maize lines was explained on the basis of hormonal parameters (PC1 component), with a lower influence of the physical parameters (PC2 component). Under such conditions, the response of lines Mo18W, B97, Ky21 and IL14h was highly dependent of Lo and correlated negatively with the SA accumulation in sap, while the response of lines Tzi8 and CML52 was highly dependent of Lh and correlated negatively with IAA levels in sap. On the other hand, the responses of lines NC350 and NC358 was closely dependent of levels of ABA and jasmonates in sap and of Gs. The response of lines Ms71 and B73 depended to a lesser extent on Lo and Lh and correlated negatively with the levels of IAA and SA, respectively. The response of line Tx303 was highly dependent on the levels of SA (Figure 5A,B) and correlated negatively with Lo values. 

Thus, for a subsequent experiment with an AM fungus, we selected a group of lines with a positive response to drought in terms of root hydraulic conductivity and growth (Ky21, Tzi8, Mo18W) and a group of lines with a negative response to drought in terms of root hydraulic conductivity and growth (Ms71, B73, NC350). 

### 2.2. Experiment 2

#### 2.2.1. AM Root Colonization and Shoot Dry Weight

As expected, non-inoculated plants did not exhibit AM root colonization. For the inoculated treatments, the percentage of mycorrhizal root length was not affected by the drought stress imposed. Hence, all inoculated maize lines showed similar values under well-watered than under drought stress conditions (data shown in Appendix A). Line Ky21 reached the highest AM root colonization length (over 60%). B73 plants reached over 55% of mycorrhizal root length and line NC350 over 50% of mycorrhizal root length. The lowest AM colonization rates were found in lines Tzi8, Ms71 and Mo18W, which had a similar AM root colonization length, ranging 40% to 44% (Appendix A).

The highest SDW values were obtained under well-watered conditions in line Ms71 inoculated with the AM fungus, as well as, in lines Ky21 and Tzi8, regardless of their mycorrhizal inoculation. On the contrary, line Mo18W exhibited the lowest growth, especially in the absence of AM inoculation (Figure 6). Line Ms71 enhanced its SDW by 31% due to AM inoculation when cultivated under well-watered conditions but was unaffected by the AM symbiosis under drought conditions. Lines B73 and Mo18W enhanced the shoot biomass production in response to AM inoculation, both under well-watered and under drought stress conditions. On the contrary, line Ky21, decreased slightly its shoot biomass production after inoculation with the AM fungus (by 7% under well-watered conditions or by 4% under drought stress). Lines NC350 and Tzi8 resulted unaffected by AM inoculation in shoot biomass. 

#### 2.2.2. Stomatal Conductance (Gs) and Efficiency of Photosystem II

The stomatal conductance values varied notably among maize lines and treatments and was reduced significantly by the drought stress imposed, except in lines Ky21, NC350 and Tzi8 when inoculated with the AM fungus (Figure 7A). B73 maize plants showed the highest decreases in Gs values due to drought stress, regardless of AM inoculation. Ky21 and Tzi8 also showed important decreases in Gs values due to drought when remained uninoculated, but not when inoculated with the AM fungus. Lines Ms71 and Mo18W showed a lower decrease in Gs values due to drought, regardless of AM inoculation. 

Regarding the efficiency of photosystem II, our hypothesis was that the AM symbiosis can help maize plants to protect their photosynthetic machinery under drought stress. Indeed, drought leads to a decrease in crop production due to inhibition of photosynthetic processes. The lowering of the photosynthetic rate caused by drought stress can induce an over-reduction in the reaction centres in and this may damage the photosynthetic machinery if the plant is unable to dissipate the excess energy [14]. Results of efficiency of photosystem II (Figure 7B) showed that for non-AM plants, drought decreased this value in all maize lines. However, for lines Ky21, NC350 and Tzi8, the AM symbiosis kept values of efficiency of photosystem II under drought stress conditions similar to values well-watered conditions. For lines B73, Ms71 and Mo18W the AM symbiosis did not maintain such values. 

#### 2.2.3. Osmotic Root Hydraulic Conductivity (Lo)

The Lo values varied notably among maize lines and mycorrhizal or drought treatments applied (Figure 8). Thus, line Ky21 exhibited the highest Lo values that were further enhanced by mycorrhization, especially under drought stress conditions (an increase by 234%). NC350 plants also notably enhanced Lo values (by 348%) as a consequence of AM inoculation. Finally, B73 and Mo18W plans also enhanced Lo values after AM inoculation, both under well-watered and under drought stress conditions. On the other hand, Tzi8 plants showed a low variation in Lo values and only increased this parameter under drought stress (by 66%), but not under well-watered conditions, while Ms71 plants only enhanced Lo by AM inoculation under well-watered conditions (by 53%), but not under drought stress. 

#### 2.2.4. Hydrostatic Root Hydraulic Conductivity (Lh)

It is remarkable that mycorrhization of lines NC350 and Tzi8 enhanced Lh values (by 101% and 81%, respectively) when subjected to drought stress (Figure 9). Non-AM Tzi8 and Mo18W plants also enhanced Lh under drought stress (by 54% and 121%, respectively). Ms71 plants did not show an alteration of Lh values either as a consequence of AM inoculation or as a consequence of water regime imposed. Finally, B73 and Mo18W plants responded negatively to AM inoculation decreasing Lh (by 35% and 45%, respectively) when subjected to drought stress conditions. 

#### 2.2.5. Principal Component Analysis

In this study, we also analysed root aquaporin gene expression and sap hormonal content. The different aquaporin genes (*ZmPIP*1;1, *ZmPIP*1;3, *ZmPIP*2;2, *ZmPIP*2;4, *ZmTIP*1;1, *ZmTIP*2;3, *ZmTIP*4;1 and *ZmNIP*2;1) and hormones (ABA, IAA, SA, JA and JA-Ile) analysed showed a high variability among treatments (See Appendix A). We conducted a principal component analysis to disentangle the maize variability in hydraulic and hormonal responses to drought stress and the implication of AM symbiosis and aquaporins in these responses. 

Results from PCA analysis are shown in Figure 10. Under well-watered conditions the mycorrhization did not alter substantially the behaviour of the different maize lines (Figure 10C vs. Figure 10A), being the aquaporin gene expression the component with the highest influence on the physiological behaviour of these lines (Figure 10B,D). In the absence of mycorrhization, the drought stress (Figure 10E vs. Figure 10A) emphasized more the particular behaviour of each line (except for B73, which exhibited a similar pattern). In the absence of mycorrhization, the behaviour of each line was mainly controlled by aquaporins and the hormones IAA and SA, and to a lesser extent by Lo and jasmonates (Figure 10B,F). Under drought stress conditions, both the mycorrhization (Figure 10G vs. Figure 10E) and the own drought stress (Figure 10G vs. Figure 10C) notably affected the responses of lines Ky21, Ms71, NC350 and B73. Indeed, under drought stress the parameters with the highest influence on the behaviour of these maize lines were the aquaporin gene expression and the levels of IAA, SA and jasmonates (Figure 10F,H). On the contrary, under drought stress Lo and Lh had little influence on the physiological behaviour of the different lines (Figure 10F,H).

## 3. Discussion

### 3.1. Importance of the AM Symbiosis and Aquaporins for Maize under Drought Stress 

Although the amount of information about plant responses to water deficit is rising continuously, our knowledge about the drought tolerance mechanisms in maize seedlings is still incomplete [4]. Maize is a primary food crop. Since 2012, it has surpassed other cereals such as wheat or rice [4]. The impact of drought on the productivity of these cereals will become of capital importance, since these three crops represent the 50% of total consumed calories in most populated earth areas [41]. The symbiosis of AM fungi with plant roots has been shown to increase the tolerance to water stress episodes in different plant species, including maize [24,42,43,44,45,46]. Thus, it has been reported that AM-plant association leads to better plant osmotic regulation, root hydraulic properties and antioxidant capacity [47,48]. In addition, AM plants generally present enhanced chlorophyll fluorescence parameters, a higher level of photosynthetic pigments and net photosynthetic rate [49], together with altered hormonal levels as compared to control plants [34,50,51].

In the case of maize plants subjected to drought stress, the improvement of its physiology by AM fungi has been attributed to a reduced oxidative damage, a better uptake of soil nutrients and water, enhanced root water transport capacity and a facilitated switching between cell-to-cell and apoplastic radial water transport pathways [40,44,52,53]. Moreover, the establishment of the AM symbiosis can modify in the host plant the pattern of membrane proteins involved in the exchange of nutrients and water between both symbiotic partners, as is the case of aquaporins [23,24,54]. Aquaporins belong to the membrane intrinsic proteins group and are located in different cell membranes. Most higher plants contain a highly diverse aquaporin family, with at least 30 isoforms. All aquaporins have the capacity to transport water to a higher or lesser extent, but some of them can also transport other molecules of high importance for plant physiology such as urea or ammonia, CO_2_, silicon, boron, hydrogen peroxide or oxygen [21,22,55]. Thus, it has been proposed that the concerted action of different aquaporin isoforms contribute to several physiological plant functions. Hence, as their possible functions in plant physiology and development are so wide, there is a need for further characterization of aquaporins in response to different environmental conditions [55,56,57]. In any case, so far there is no doubt that they play an important role in water distribution at the whole plant level and in plant responses to different stresses [18,21,56,57]. In this regard, it is known that when transpiration is restricted due to drought stress, the predominant pathway for radial water movement is the cell-to-cell path [58] and it has been found that aquaporins regulate water flow through such pathway [13,19,44,45]. In AM maize plants, the expression of these proteins varies according to the severity of the stress and depends on the duration of the water shortage period [24]. Moreover, it has been shown that the presence of the AM fungus in the root increases the water permeability of root cells, related to induction of some aquaporin genes and increase in the phosphorylation status of PIP2s, which implies a higher activity of their water channels [45]. 

### 3.2. Maize Physiological and Hydraulic Responses to Drought and Mycorrhization

Many of the physiological responses of plants to drought stress are directed toward the control of root hydraulic conductivity, osmotic adjustment and transpiration [59]. Stomatal closure and reduced transpiration are conserved mechanisms in all maize genotypes studied, regardless of AM inoculation. Augé et al. [60] conducted a meta-analysis of 460 studies and revealed that, even if AM-inoculated C3 plants usually show higher Gs values, C4 plants featured increases in Gs of around 12%. In this study, AM symbiosis increased Gs in B73, NC350 and Tzi8 genotypes. In contrast, no differences were found in Gs values Ky21, Ms71 and Mo18W genotype subjected to water stress. This could probably be related to the larger shoot biomass of Ky21 and Ms71 plans with the subsequent increased transpiring area, or to the fact that drought-sensitive plants had generally lower Gs values than the drought-tolerant ones. 

When plants grow under non-stressful conditions the radial water flow is mainly apoplastic, with transpiration as the driven force. However, when transpiration is restricted due to stressful conditions such as drought, water mainly moves by the cell-to-cell pathway, governed by the osmotic gradient between soil solution and xylem sap. Thus, under drought conditions, root hydraulics is adjusted by switching between both pathways [61] and the relative contribution of both pathways to overall root water uptake may change considerably [62]. The osmotic root hydraulic conductivity (Lo) estimates water movement via the cell-to-cell pathway and is highly related to the activity of aquaporins in the plasma membrane [63]. When plants are exposed to water deprivation, a reduction in Lo is usually reported [59,64], probably as a mechanism for preventing water loss. Such reduction was clear for B73, Ms71 and Mo18W plants. In contrast, Ky21, NC350 and Tzi8 plants did not reduce Lo values due to drought. It is noteworthy that under drought stress conditions, AM symbiosis increased Lo as compared to control plants in all genotypes, except in Ms71 plants. This enhancement is in accordance with previous studies on AM plants subjected to drought [6,24,34,44,45,65]. The reported increase in Lo in AM plants could be related to an increased expression of plant aquaporins, but also of the own fungal aquaporins [34,45]. Additionally, the increase in Lo in AM plants may be due to post-translational modifications of aquaporins such as changes in their phosphorylation status that enhance their water channel activity [56] Moreover, changes in size or density of plasmodesmata in AM roots [66] could contribute to enhanced Lo values in these plants. Indeed, it has been demonstrated that symplastic movement of water via plasmodesmata can significantly contribute to Lo values [67]. Moreover, the regulation of aquaporin activity in response to osmotic stresses involve post-transcriptional changes, including the formation of heterotetramers [68], which improve the water channel activity [69,70]. 

In the context of the above information, it is clear that the AM symbiosis is particularly important to increase root water transport during periods of drought stress due to its ability to modulate plant aquaporins. Indeed, under such conditions the formation of apoplastic barriers increases in root endodermis in order to avoid water losses [71]. The presence and activity of aquaporin in root cell membranes serves to circumvent these barriers and benefits from an alternative water supply at the interface with mycelial membranes [46]. Thus, in drought-stressed roots where radial water flow is largely impeded by apoplastic barriers, the transcellular pathway regulated by the AM symbiosis and by aquaporins would be of particular importance for hydraulic adjustment. Indeed, our previous studies already suggested a higher flexibility of AM plants to switch between apoplastic and transcellular water transport pathways according to the water demands of the shoot, allowing for a more efficient response to water scarcity [44,52].

Aquaporin abundance in root cortex cells may alter Lo, especially during water shortage [17], when aquaporins are thought to be regulated for the maintenance of the adequate water balance [72,73] Among the different aquaporin types, PIPs were proved to contribute to the adaptation of plants to drought episodes, also contributing to rehydration of the whole plant after water shortage [74] Moreover, transcriptomic analysis of drought tolerant and sensitive RILs in maize suggested that down-regulation of aquaporins is a mechanism contributing to the drought tolerance by upholding tissue turgor over longer time than drought-sensitive lines [4].

To understand the importance of aquaporins regulation in maize plants, it must be taken into account that plant aquaporins not only transport water, but also many other substrates having a physiological role such as urea, glycerol, boric acid, silicic acid, hydrogen peroxide or gaseous molecules such as carbon dioxide, ammonia or oxygen [22,55,75]. As the substrates that can be transported by the plant aquaporins are highly diverse, the aquaporin isoforms regulated by the AM symbiosis may have a role in regulation of nutrient uptake and translocation along plant tissues, carbon fixation, leaf and root hydraulics, stomatal movement or signalling processes. In this context, according to the PCA analysis performed (Figure 10), the eight maize aquaporin genes studied here (*ZmPIP*1;1, *ZmPIP*1;3, *ZmPIP*2;2, *ZmPIP*2;4, *ZmTIP*1;1, *ZmTIP*2;3, *ZmTIP*4;1 and *ZmNIP*2;1) seem to make a significant contribution to the responses of the different maize lines under the different environmental conditions of our experiments. These aquaporin genes were selected in previous studies since they were regulated by the AM symbiosis under different conditions of drought stress (moderate or severe) and in different varieties of maize (drought-sensitive and drought-tolerant) [6,24]. The aquaporins ZmPIP2;2 and ZmPIP2;4 have been shown to exhibit a high-water transport capacity, while ZmPIP1;1 and ZmPIP1;3 have low capacity for water transport by themselves [24], but they could interact with PIP2s to increase their water transport capacity [69,70]. Hence, all these PIPs can improve root hydraulic properties. ZmTIP1;1, ZmTIP2;3, ZmTIP4;1 and ZmNIP2;1 can transport water, but their Pf values are low in comparison to ZmPIP2;2 and ZmPIP2;4 [24]. Thus, these aquaporins must be affecting the maize physiology by other mechanisms. For instance, ZmTIP1;1 and ZmNIP2;1 can transport ammonium and/or urea [6,24,26], which opens the door to their participation in the mobilization of nitrogen compounds to be used by the plant under stressful conditions. Almost all of the aquaporins studied here could transport hydrogen peroxide, especially ZmTIP1;1 [24]. Thus, they could also play a role in the detoxification of this reactive oxygen species or in signalling events mediated by H_2_O_2_ and contribute in such way to the plant drought tolerance.

### 3.3. Maize Hormonal Responses to Drought and Mycorrhization

Among plant responses to stresses, ABA is considered the most important signal transduction pathway [76,77]. ABA biosynthesis is one of the fastest plant responses to drought stress. However, extensive interactions between the different hormonal signalling pathways exists in plants [33,78,79]. Therefore, the signalling responses of ABA, auxins and jasmonates are closely interconnected and play an important role in plant physiology during osmotic stress [79]. Thus, the role of plant hormones in the regulation of root water transport has been studied. For instance, MeJA has been shown to increase Lo in different plant species (*Phaseolus vulgaris*, *Arabidopsis thaliana* and *Solanum lycopersicum*) in a calcium- and ABA-dependent way [34,80]. ABA generally also enhances Lo values [50,51,81]. Moreover, the combined effects of ABA and AM symbiosis on root water transport capacity and aquaporins regulation have been reported [50,51,82,83]. On the contrary, both IAA and SA decrease Lo values [25,84,85]. In the case of IAA, its role as a regulator of root water transport is not well understood [86]. However, the existence of a negative effect of IAA on the internal cell component of root water conductivity (Lo) suggests that aquaporins are involved in the IAA-dependent inhibition of this internal cell water pathway [40]. Indeed, Péret et al. [85] has shown that a treatment with auxin inhibited aquaporin genes through the auxin response factor ARF7 and this consequently reduced hydraulic conductivity both at cell and whole root levels. The inhibition of Lo by SA has been related to the internalization of PIPs in cell vesicles [84] or to fine regulation in roots of the aquaporins ZmPIP2;4 and ZmTIP1;1, which have a high-water transport capacity [25]. Alternatively, Du et al. [87] found that SA could also control aquaporin activity by affecting their abundance in the plasmalemma.

In this study, the responses of the different maize NAM lines (experiment 1) were highly governed by their hormonal content. Indeed, when plants were exposed to drought stress conditions most of the variability among maize NAM lines was explained on the basis of hormonal parameters (PC1 component). Under such conditions, the response of lines Mo18W, B97, Ky21, B73 and IL14h correlated negatively with the SA accumulation in sap, while the response of lines Tzi8, Ms71 and CML52 correlated negatively with IAA levels in sap. On the other hand, the responses of lines NC350 and NC358 was closely dependent of levels of ABA and jasmonates in sap. When plants were inoculated with the AM fungus (experiment 2) both the mycorrhization and the own drought stress notably affected the responses of lines Ky21, Ms71, NC350 and B73, being the aquaporin gene expression and the sap levels of IAA, SA and jasmonates the parameters with the highest influence on the behaviour of these maize lines.

In conclusion, results from this study show that plant aquaporins and hormones are highly involved in the physiological responses of maize plants to drought stress and both are altered by the AM symbiosis. The aquaporins regulated by the AM symbiosis such as ZmPIP2;2, ZmPIP2;4, ZmTIP1;1, ZmTIP2;3, ZmTIP4,1 and ZmNIP2;1 have a significant capacity for water transport across cell membranes and, thus, the symbiosis can alter the radial water transport capacity in the host plant. Other aquaporin isoforms such as ZmPIP1;1 and ZmPIP1;3 can increase the water transport capacity of other isoforms by heterotetramers formation [69,70]. In addition, ZmTIP1;1 and ZmNIP2;1 could participate in the mobilization of nitrogen compounds to be used by the plant under stressful conditions. Moreover, several of the aquaporins studied here can transport hydrogen peroxide. Thus, they could also play a role in the detoxification of this reactive oxygen species or in signalling events, contributing in such way to the plant drought tolerance. Hormones such as IAA, SA, ABA and jasmonates must be involved in this process either by regulating the own plant-AM fungus interaction, the activity of the aquaporins or by inducing posttranscriptional changes in these aquaporins, which in turns alter their water transport activity. Indeed, an intricate relationship between Lo, aquaporins and phytohormones has been observed, revealing a complex network controlling water transport in maize roots.

## 4. Materials and Methods

### 4.1. Design of the Experiments and Statistical Analysis

#### 4.1.1. Experiment 1

The experiment consisted of a factorial design with two factors: (1) maize NAM line, so that we tested the 27 inbred parents of the maize NAM population, (2) stress treatment, so that half of the plants was cultivated under well-watered conditions (WW) throughout the entire experiment and the other half of the plants was subjected to drought stress (D) for 15 days before harvest. The different combinations of these factors provided a total of 54 treatments. Twelve replicates were used for each treatment, providing a total of 648 plants.

The SPSS Statistics (Version 27, IBM Analytics, IBM Corp., New York, NY, USA) was used to perform data analysis. Data were subjected to analysis of variance (ANOVA) with maize line, water regime and maize line-water regime interaction as sources of variation. Post-hoc comparisons with the Duncan’s test were used to find out differences between groups. Within each maize line a Student’s *t*-test was also applied for pairwise comparison of data between well-watered and droughted treatments.

A principal component analysis (PCA) was performed using the built-in R functions prcomp and PCA.

#### 4.1.2. Experiment 2

The experiment consisted of a factorial design with three factors: (1) Maize NAM lines, so that we tested 6 maize lines with contrasting responses to drought in terms of osmotic root hydraulic conductivity (Lo) and growth (as determined in Experiment 1); (2) microbial inoculation treatment, with non-inoculated control plants (C) or plants inoculated with the AM fungus *Rhizophagus irregularis* (AM); (3) stress treatment, so that one half of the plants was cultivated under well-watered conditions (WW) throughout the entire experiment and the other half of the plants was subjected to drought stresses (D) for 15 days before harvest. The different combinations of these factors gave a total of 24 treatments. Fifteen replicates were used for each treatment, giving a total of 360 plants.

The SPSS Statistics (Version 26, IBM Analytics) was used to perform data analysis. Within each maize line, data were subjected to analysis of variance (ANOVA) with inoculation treatment, water regime and inoculation-water regime interaction as sources of variation. Post-hoc comparisons with the Duncan’s test were used to find out differences between groups.

A principal component analysis (PCA) was performed using the built-in R functions prcomp and PCA.

### 4.2. Soil and Biological Materials

The soil used in both experiments was a loam and was collected at the grounds of IFAPA (Granada, Spain), sieved (2 mm), diluted with quartz-sand (<1 mm) (1:1, soil:sand, *v*/*v*) and sterilized by steaming (100 °C for 1 h on 3 consecutive days). The soil pH was 8.1 (water) and contained 0.85% organic matter, nutrient concentrations (mg kg^−1^): N, 1; P, 10 (NaHCO3-extractable P); K, 110. The soil texture mas made of 38.3% sand, 47.1% silt and 14.6% clay.

The maize seeds used in both experiments were available through the Maize Genetics Cooperation Stock Center (MGCSC; maizecoop.cropsci.uiuc.edu/, accessed on 5 July 2022) and GRIN-Global (npgsweb.ars-grin.gov/gringlobal/search, accessed on 5 July 2022). The MGCSC is operated by USDA/ARS, located at the University of Illinois and integrated with the National Plant Germplasm System (NPGS). They serve the maize research community by collecting, maintaining and distributing seeds of maize genetic stocks.

For experiment 1, the twenty-seven parent lines used were: B73 (PI 550473); B97 (PI 564682); CML52 (PI 595561); CML69 (Ames 28184); CML103 (Ames 27081); CML228 (Ames 27088); CML247 (PI 595541); CML277 (PI 595550); CML322 (Ames 27096); CML333 (Ames 27101); HP301 (PI 587131); Il14H (Ames 27118); Ki3 (Ames 27123); Ki11 (Ames 27124); Ky21 (Ames 27130); M37W (Ames 27133); M162W (Ames 27134); Mo18W (PI 550441); Ms71 (PI 587137); Mo17 (PI 558532); NC350 (Ames 27171); NC358 (Ames 27175); Oh7B (Ames 19323); Oh43 (Ames 19288); P39 (Ames 28186); Tx303 (Ames 19327) and Tzi8 (PI 506246).

For experiment 2, the six parent lines used were Ky21 (Ames 27130), Tzi8 (PI 506246) and Mo18W (PI 550441), which exhibited a good response to drought in terms of osmotic root hydraulic conductivity (Lo) and growth and B73 (PI 550473), Ms71 (PI 587137), and NC350 (Ames 27171), which exhibited a poor response to drought in terms of Lo and growth.

Mycorrhizal inoculum was bulked in an open-pot culture of *Z. mays* L. at the Zaidin Experimental Station (EEZ) Collection, and consisted of soil, spores, mycelia and infected root fragments. The AM fungus was *Rhizophagus irregularis* (synonymous of *Glomus intraradices*, Schenck and Smith), strain EEZ 58. This AM fungal isolate has been used since long time in our experiments as it has shown a high efficiency to improve plant tolerance to drought stress with different host plants species [6,13,24,25,26,27,50,51,52,53,65]. Ten grams of inoculum with about 60 infective propagules per gram (according to the most probable number test), were added to appropriate pots at sowing time. Non-inoculated control plants received the same amount of autoclaved mycorrhizal inoculum together with a 3 mL aliquot of a filtrate (<20 µm) of the AM inoculum in order to provide a general microbial population free of AM propagules.

### 4.3. Growth Conditions

Both experiments were conducted in consecutive seasons in a greenhouse with the following conditions: 16/8 h light/dark period, a relative humidity of 50–60% and temperatures of 19/24 °C (night/day). The average photosynthetic photon flux density was 800 µmol m^−2^ s^−1^, as measured with a light meter (LICOR, Lincoln, NE, USA, model LI-188B). Plants were cultivated for a total of 8 weeks and, 4 weeks after sowing, all plants started receiving 10 mL per pot and per week of Hoagland nutrient solution [88] containing only 25% of P, in order to provide basic nutrients, but avoiding inhibition of AM symbiosis due to a high P application.

The soil moisture was controlled with a ML2 ThetaProbe (AT Delta-T Devices Ltd., Cambridge, UK). Thus, during the first 6 weeks after sowing, water was daily supplied to maintain soil at 100% of field capacity in all treatments. A previous experiment using a pressure plate apparatus showed that the 100% soil water holding capacity corresponds to 22% volumetric soil moisture measured with the ThetaProbe. Then, half of the plants (unstressed plants) were maintained under the above-mentioned conditions during the entire experiment. The other half of the plants (stressed plants) were allowed to dry until soil water content reached 60% of field capacity (one day needed). The 60% of soil water holding capacity corresponds to 7% volumetric soil moisture measured with the ThetaProbe (also determined previously with a pressure plate apparatus). The soil water content was daily measured with the ThetaProbe ML2 before rewatering (at the end of the afternoon), reaching a minimum soil water content around 55% of field capacity in the drought stressed treatments. The amount of water lost was daily replaced to each pot in order to keep the soil water content at the desired levels of either 7% (stressed plants) or 22% (non-stressed plants) of volumetric soil moisture [89]. Plants were maintained under such conditions for 15 additional days before harvesting.

### 4.4. Measurements

#### 4.4.1. Biomass Production

At harvest (8 weeks after sowing), the shoot and root system of twelve replicates per treatment were separated and the dry weight (DW) measured after drying in a forced hot-air oven at 70 °C for 2 days.

#### 4.4.2. Symbiotic Development

In experiment 2, the percentage of mycorrhizal root colonization was estimated by visual observation according to Phillips and Hayman [90] and the extent of mycorrhizal colonization was quantified according to the gridline intersect method [91] in five replicates per treatment.

#### 4.4.3. Stomatal Conductance

Stomatal conductance measurements were taken in the second youngest leaf from ten different plants of each treatment. Stomatal conductance was measured two hours after the onset of photoperiod with a porometer system (Porometer AP4, Delta-T Devices Ltd., Cambridge, UK) following the user manual instructions.

#### 4.4.4. Photosynthetic Efficiency

The efficiency of photosystem II was measured with a FluorPen FP100 (Photon Systems Instruments, Brno, Czech Republic), which allows for a non-invasive assessment of plant photosynthetic performance by measuring chlorophyll a fluorescence. FluorPen quantifies the quantum yield of photosystem II as the ratio between the actual fluorescence yield in the light-adapted state (FV’) and the maximum fluorescence yield in the light-adapted state (FM’), according to Oxborough and Baker [92]. Measurements were taken in the second youngest leaf of ten different plants of each treatment.

#### 4.4.5. Osmotic Root Hydraulic Conductivity (Lo)

Before harvest, the osmotic root hydraulic conductivity (Lo) was measured on detached roots exuding under atmospheric pressure for two hours [13]. Under these conditions, water is only moving following an osmotic gradient. Therefore, the water would be moving through the cell-to-cell path [58]. Eight plants per treatment were used for this determination. Lo was calculated as Lo = Jv/∆Ψ, where Jv is the exuded sap flow rate and ∆Ψ is the osmotic potential difference between the exuded sap and the nutrient solution where the pots were immersed. These measurements were carried out 3 h after the onset of light.

#### 4.4.6. Hydrostatic Root Hydraulic Conductivity (Lh)

Lh was determined at noon in seven plants per treatment with a Scholander pressure chamber as described by Bárzana et al. [52]. A gradual increase in pressure (0.3, 0.4 and 0.5 MPa) was applied at 2-min intervals to the detached roots. Sap was collected at the three pressure points. Sap flow was plotted against pressure, with the slope being the root hydraulic conductance (L) value. Lh was determined by dividing L by root dry weight (RDW) and expressed as mg H_2_O g RDW^−1^ MPa^−1^ h^−1^.

#### 4.4.7. Sap Hormonal Content

ABA, IAA, SA, JA and JA-Ile were analysed in the sap collected for Lo measurement according to Albacete et al. [93] with some modifications. Briefly, xylem sap samples were filtered through 13 mm diameter Millex filters with 0.22 µm pore size nylon membrane (Millipore, Bedford, MA, USA). The deuterium-labelled internal standard used for hormones determination were the following: ^2^H_5_-Indole-3-Acetic acid (D-IAA), ^2^H_6_-(+)-cis,trans-Abscisic acid (D-ABA), ^2^H_2_-N-(-)-Jasmonoyl Isoleucine (D-JA-Ile) and ^2^H_4_-Salicylic acid (D-SA), obtained from OlChemin Ltd. (Olomouc, Czech Republic). The ^2^H_5_-Jasmonic acid (D-JA) was obtained from CDN Isotopes (Pointe-Claire, QC, Canada). Ten µL of each internal standard was added to the filtrate. Subsequently, 10 µL of filtrate were injected in a U-HPLC-MS system consisting of an Accela U-HPLC system (ThermoFisher Scientific, Waltham, MA, USA) coupled to an Exactive^TM^ mass spectrometer (ThermoFisher Scientific) using a heated electrospray ionization (HESI) interface. Mass spectra were obtained using Xcalibur software version 2.2 (ThermoFisher Scientific, Waltham, MA, USA). For quantification of the plant hormones, calibration curves were constructed for each analysed component (1, 10, 50, and 100 µg L^−1^).

#### 4.4.8. Quantitative Real-Time RT-PCR

In experiment 2, three biological replicates of maize roots were used to extract total RNA as described in Quiroga et al. [6]. First-strand cDNA was synthesized using 1 µg of purified RNA with the Maxima H Minus first strand cDNA synthesis kit (Thermo Scientific^TM^, Waltham, MA, USA), following the manufacturers’ instructions.

The expression of a group of eight maize aquaporin genes (*ZmPIP*1;1, *ZmPIP*1;3, *ZmPIP*2;2, *ZmPIP*2;4, *ZmTIP*1;1, *ZmTIP*2;3, *ZmTIP*4;1 and *ZmNIP*2;1) selected in previous studies [6] was measured by qRT-PCR using 1 µL of diluted cDNA (1:9) with Applied Biosystems^TM^ PowerUp^TM^ SYBR^TM^ Green Master Mix in a QuantStudio^TM^ 3 system (Applied Biosystems, Waltham, MA, USA). Four reference genes were measured in all the treatments for normalization of gene expression values. These genes were polyubiquitin (accession number gi: 248338), tubulin (gi: 450292), GAPDH (gi: 22237) and elongation factor 1α (gi: 2282583) [24]. “NormFinder” algorithm [94] (https://moma.dk/normfinder-software, accessed on 5 July 2022) was used to select two best-performing reference genes under our specific conditions and the standardization was carried out based on the expression of these two genes. Thus, expression levels were normalized according to *Zmtubulin* and *ZmGAPDH* genes. The relative abundance of transcripts was calculated using the 2^−∆∆Ct.^ method [95]. The threshold cycle (Ct) of each biological sample was determined in duplicate. Negative controls without cDNA were used in all PCR reactions.

## Figures and Tables

**Figure 1 ijms-23-09822-f001:**
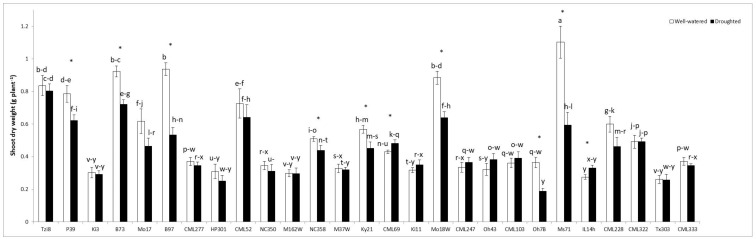
Shoot dry weight in the whole maize NAM population cultivated under well-watered conditions or subjected to drought stress. Data represents the means of 10 values ± S.E. Different letters indicate significant differences between treatments (*p* < 0.05) based on Duncan’s test. Asterisks indicate significant differences between well-watered and droughted treatments within each maize line, according to Student’s *t*-test.

**Figure 2 ijms-23-09822-f002:**
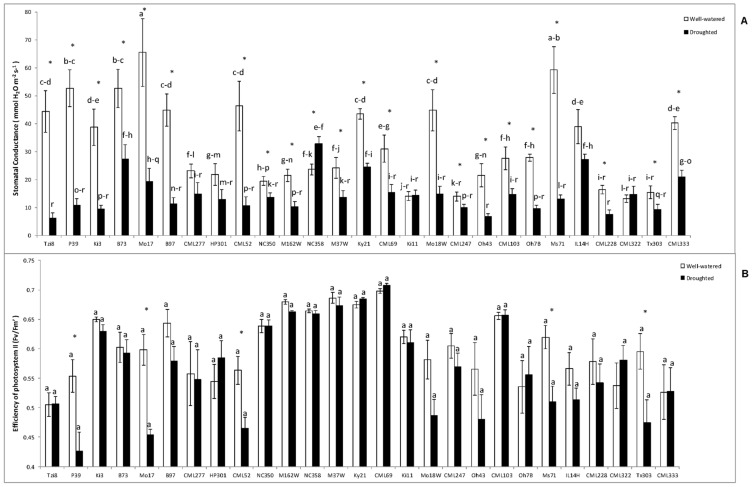
(**A**) Stomatal conductance (gs) and (**B**) photosystem II efficiency in the light-adapted state (∆Fv/Fm’) in the whole maize NAM population cultivated under well-watered conditions or subjected to drought stress. Data represents the means of 10 values ± S.E. Different letters indicate significant differences between treatments (*p* < 0.05) based on Duncan’s test. Asterisks indicate significant differences between well-watered and droughted treatments within each maize line, according to Student’s *t*-test.

**Figure 3 ijms-23-09822-f003:**
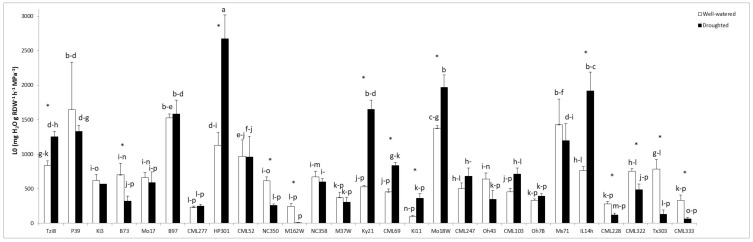
Osmotic root hydraulic conductivity (Lo) in the whole maize NAM population cultivated under well-watered conditions or subjected to drought stress. Data represents the means of 6 values ± S.E. Different letters indicate significant differences between treatments (*p* < 0.05) based on Duncan’s test. Asterisks indicate significant differences between well-watered and droughted treatments within each maize line, according to Student’s *t*-test.

**Figure 4 ijms-23-09822-f004:**
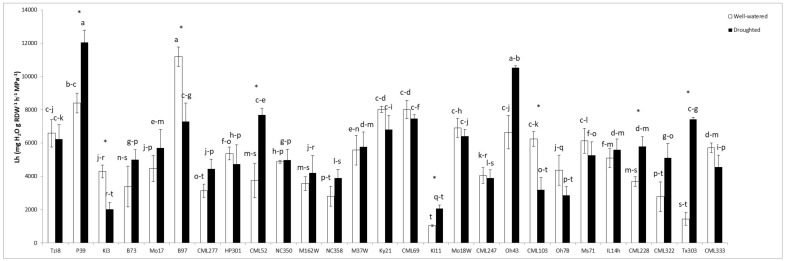
Hydrostatic root hydraulic conductivity (Lh) in the whole maize NAM population cultivated under well-watered conditions or subjected to drought stress. Data represents the means of 6 values ± S.E. Different letters indicate significant differences between treatments (*p* < 0.05) based on Duncan’s test. Asterisks indicate significant differences between well-watered and droughted treatments within each maize line, according to Student’s *t*-test.

**Figure 5 ijms-23-09822-f005:**
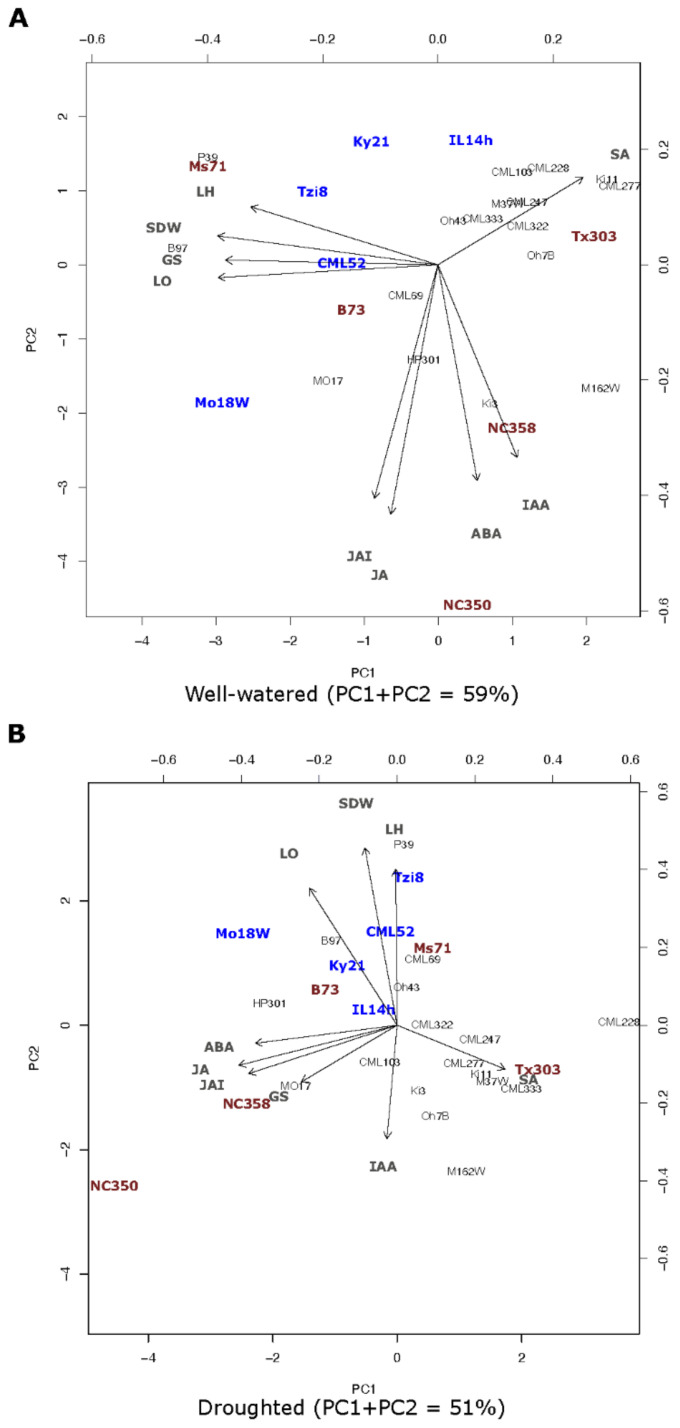
PCA analysis with data for shoot dry weight, Gs, Lo, Lh and sap hormonal content (ABA, IAA, SA, JA and JA-Ile) in the whole maize NAM population cultivated under well-watered conditions (**A**) or subjected to drought stress (**B**).

**Figure 6 ijms-23-09822-f006:**
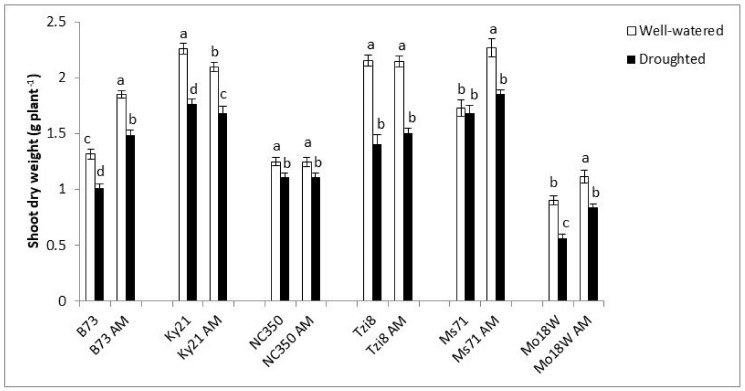
Shoot dry weight in six maize NAM lines (B73, Ky21, NC350, Tzi8, Ms71 and Mo18W) inoculated or not with an arbuscular mycorrhizal (AM) fungus. Plants were cultivated under well-watered conditions or subjected to drought stress. Data represents the means of 15 values ± S.E. Within each maize line, different letters indicate significant differences between treatments (*p* < 0.05) based on Duncan’s test.

**Figure 7 ijms-23-09822-f007:**
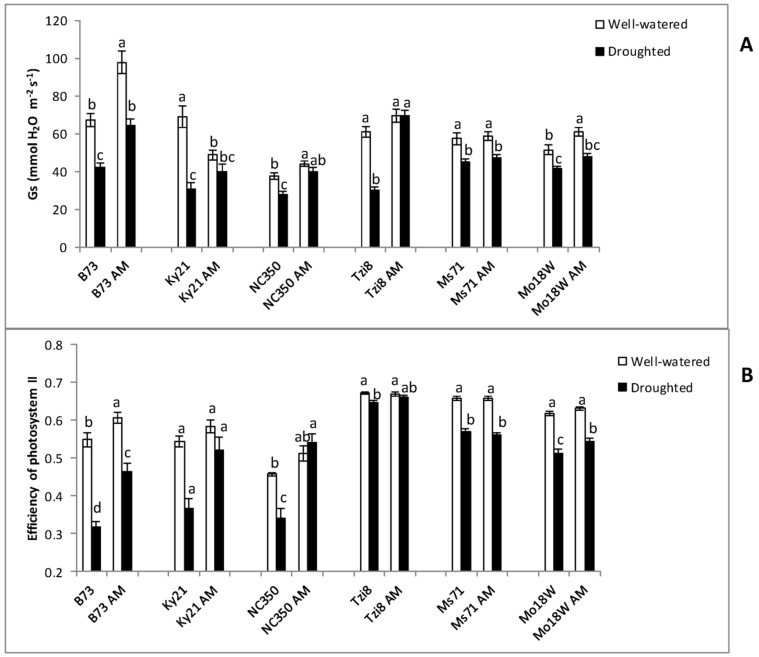
(**A**) Stomatal conductance (Gs) and (**B**) photosystem II efficiency in the light-adapted state (∆Fv/Fm’) in six maize NAM lines (B73, Ky21, NC350, Tzi8, Ms71 and Mo18W) inoculated or not with an arbuscular mycorrhizal (AM) fungus. Plants were cultivated under well-watered conditions or subjected to drought stress. Data represents the means of 10 values ± S.E. Within each maize line, different letters indicate significant differences between treatments (*p* < 0.05) based on Duncan’s test.

**Figure 8 ijms-23-09822-f008:**
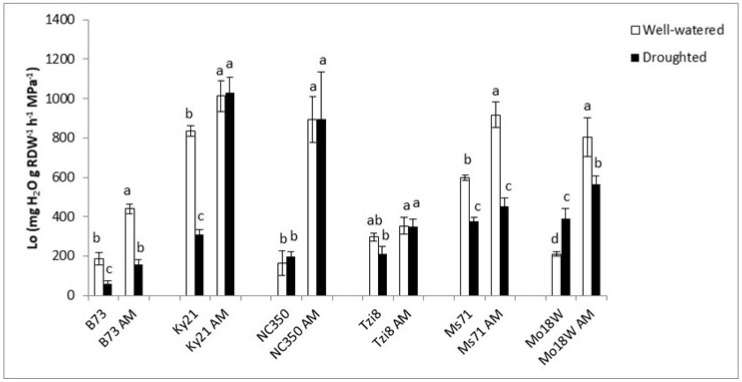
Osmotic root hydraulic conductivity (Lo) in six maize NAM lines (B73, Ky21, NC350, Tzi8, Ms71 and Mo18W) inoculated or not with an arbuscular mycorrhizal (AM) fungus. Plants were cultivated under well-watered conditions or subjected to drought stress. Data represents the means of 6 values ± S.E. Within each maize line, different letters indicate significant differences between treatments (*p* < 0.05) based on Duncan’s test.

**Figure 9 ijms-23-09822-f009:**
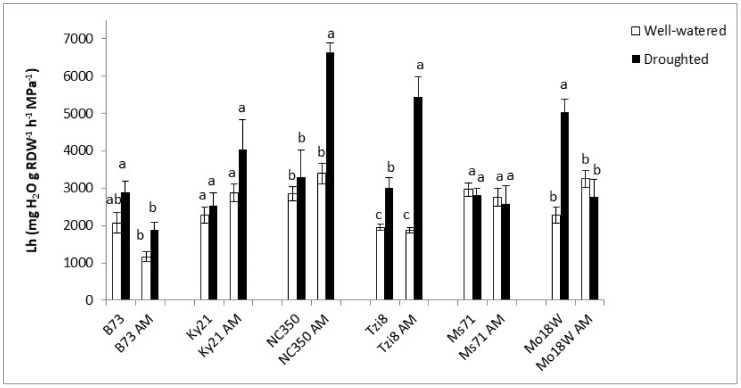
Hydrostatic root hydraulic conductivity (Lh) in six maize NAM lines (B73, Ky21, NC350, Tzi8, Ms71 and Mo18W) inoculated or not with an arbuscular mycorrhizal (AM) fungus. Plants were cultivated under well-watered conditions or subjected to drought stress. Data represents the means of 6 values ± S.E. Within each maize line, different letters indicate significant differences between treatments (*p* < 0.05) based on Duncan’s test.

**Figure 10 ijms-23-09822-f010:**
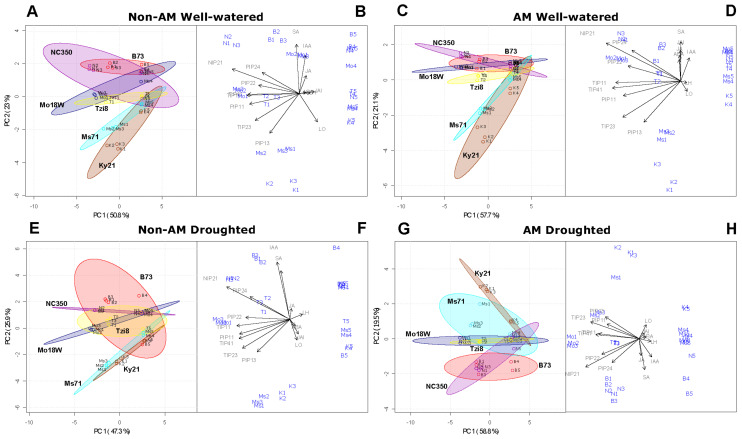
PCA analysis with data for Lo, Lh, sap hormonal content (ABA, IAA, SA, JA and JA-Ile) and expression of eight maize aquaporin genes regulated by the AM symbiosis (*ZmPIP*1;1, *ZmPIP*1;3, *ZmPIP*2;2, *ZmPIP*2;4, *ZmTIP*1;1, *ZmTIP*2;3, *ZmTIP*4;1 and *ZmNIP*2;1). The analysis was conducted in six maize NAM lines (B73, Ky21, NC350, Tzi8, Ms71 and Mo18W) inoculated (**C**,**D**,**G**,**H**) or not (**A**,**B**,**E**,**F**) with an arbuscular mycorrhizal (AM) fungus. Plants were cultivated under well-watered conditions (**A**–**D**) or subjected to drought stress (**E**–**H**). (**A**,**C**,**E**,**G**) represent the clouds that group the different data from a given maize line. (**B**,**D**,**F**,**H**) represent the contribution of each measured parameter on the behaviour of the maize lines. Points labelled as B1–B5 refer to B73 data, those labelled as K1–K5 refer to Ky21 data, those labelled as N1–N5 refer to NC350 data, those labelled as T1–T5 refer to Tzi8 data, those labelled as Ms1–Ms5 refer to Ms71 data and points labelled as Mo1–Mo5 refer to Mo18W data.

## Data Availability

Not applicable.

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
