# Peer review of "Using the Maize Nested Association Mapping (NAM) Population to Partition Arbuscular Mycorrhizal Effects on Drought Stress Tolerance into Hormonal and Hydraulic Components"

_ijms, 2022, doi:10.3390/ijms23179822_

Round 1

Reviewer 1 Report (Previous Reviewer 1)

Submitted manuscript describes effect of arbuscular mycorrhizal colonization of maize root on plant drought stress tolerance. Authors used in their research maize NAM population to screen for lines more resistant to stress and characterised selected lines by checking parameters like biomass production, stomatal conductance, photosystem II efficiency, osmotic root hydraulic conductivity, hydrostatic root hydraulic conductivity, plant hormones level and transcript level of selected genes.
General remarks:
1. Page numeration is incorrect, as well as the numeration of the lines; it should start at the beginning of the manuscript and finish at the very end while in the current version of the paper every chapter/subchapter starts with a new numeration for pages and/or lines. Improving that will make all the needed manuscript corrections much easier.
2. Colors used in the figures should be adjusted to colorblind-friendly palette
3. Graphs quality (Fig.5 & Fig.10) is not sufficient for proper data analysis, the resolution is low, and the line names are hard to read out due to color and font used by the authors.
Since the manuscript numeration is incorrect all reviewer remarks are presented in the order of the appearance in the submitted version of the paper to avoid any misunderstandings.
Detailed remarks:
1. Explanation for all the shortcuts upon first use should be included in the manuscript, thus RIL (Recombinant Inbred Line) should be explained as well (page 2 line 50).
2. Page 2 line 98 misspell; currently is “aunxins”, should be auxins.
3. Page 3 line 99 the slash sign is missing; should be or/and.
4. Page 11 subchapter 2.1.5; no statistical analysis for hormones level measurements was performed, thus its hard to determine whether observed differences are sufficient to have a meaning.
5. Page 13 line 41 misspell; is “sowed”, probably authors wanted to use “showed”.
6. Indications of the graphs (Fig.2 & 7) are not clear, A/B marks should be used on the side of the graph, not on it.
7. Fig.7 x axis of the first graph (A) should have its own indication for the presented lines at the bottom for easier and clearer understanding of the content.
8. Page 17 subchapter 2.2.5; there are no information why authors have taken into consideration those specific aquaporins. Please supply missing details either in the introduction or the results section.
9. Page 1-3, subchapter 3.2; authors are providing all the information about aquaporins but do not discuss the results obtained by them in quantitative PCR experiments.
10. Page 3 line 108; the process that authors are describing in response to drought stress is ABA biosynthesis, not synthesis; it is a multi-step process that involves number of enzymes and is conducted in a living organism, thus it covers the definition of biosynthesis pathway. Please correct this.
11. Page 3 line 123 please correct “this reduced…” to “this consequently reduced…” for proper understanding of the sentence.
12. Page 4 line 176; authors do not explain either here or throughout the rest of the manuscript why they used this specific strain of AM fungus. Please supply short description.
13. Page 6 subchapter 4.4.8; no information about standards used for hormones measurements are included, please supply information about those including manufacturer name.
14. The values of the measurements for the hormones presented as Fig. S1 and S3 differ (sometimes by few folds, sometimes the tendency is completely reversed) between first and second experiment within the same lines in well-watered and drought stress conditions (even without AM). To be exacts: ABA content – lines B73 and MO18W; IAA content – lines B73 and NC350; SA content – lines B73 and MO18W; JA content – line NC350; Ja-Ile content – lines NC350and Tzi8. Either the method used for the measurements gives not reproducible data or the growth conditions were not the same in those two experiments (although materials and methods section claims they were identical). Authors should explain that. Similarly, the issue occurs for Lo and Lh measurements for lines Ky21 (Fig.8) and Mo18W (Fig.9) respectively.
15. Authors use two different ways of writing the line name for MO18W/Mo18W and MS71/Ms71, please unify this throughout the manuscript and within the graphs.

Author Response

Please, see the attached doccument which contains answers to all comments. 

Reviewer 2 Report (New Reviewer)

The Ruiz-Lozano et al’s paper investigated how drought stress alters the radial water movement and physiology in the maize NAM population, focusing the effect of AM fungus in relation to plant aquaporins and phytohormones. The authors successfully obtained a comprehensive data for the response of drought in NAM population lines including growth, hormones levels, water transport and aquaporin levels, in relation to mycorrhizal inoculum. Although the results are rather complicated, this study will contribute greatly to the breeding of maize to cope with future climate change.  

Followings are some minor comments that will improve the MS.

1.       Fig. 1, and other bar graphs. The line names are hard to recognize. Please order the line names in an alphabetical order.

2.       Line 118 – 121. From the statistical results in figure 1, it seems that the SDW was the largest in MS71. Please check the data.

3.       Page 7, Line 5. Where is the data for biomass production?

4.       Page 13. Line 16-17. What is the meaning of “correlated negatively with all the hormones measured in sap”? Which traits was negatively correlated to hormones?

5.       Page 13, Line 40 – 46. The data for mycorrhizal root is important; the effect of mycorrhizal inoculum, e.g., the difference in mycorrhizal root length between plants with and without inoculum must be shown for all lines. I recommend the authors to show the data in text, or in a table or in a figure. What is the meaning of “B73 plants reached over 55% of mycorrhizal root length”? How did the authors recognize the mycorrhiza?

6.       Page 13, Line 47-49. This explanation is inconsistent to the data. For the inoculated plants under well-watered condition, the shoot dry weight seems to be the highest in Ms71. Please check the data and explanation carefully.

7.       Figure 7B. What is the authors’ hypothesis concerning the drought response of PSII efficiency in relation to mycorrhizal inoculum? Please explain it briefly in this part.

8.       Figure 10. The result is complicated. Please add more explanation to understand the data. What is the meaning of the colors in the figures A, E, E, and F? How can the readers recognize the effect of aquaporin?

Author Response

Please, see the attached doccument which contains the answers to all your comments. 

This manuscript is a resubmission of an earlier submission. The following is a list of the peer review reports and author responses from that submission.

Round 1

Reviewer 1 Report

Part of the discussion chapter (line 20-40) is copied from previous paper published by the authors in 2017 (https://doi.org/10.3389/fpls.2017.01056 ; from introduction part, page 2, second column, aquaporins description) with barely any/no changes, thus it is considered as a plagiarism.